# Efficacy and safety of vapocoolant spray for vascular puncture in children and adults: A systematic review and meta-analysis

**Lan Wang**[1☯], **Liu Fang**[1☯], **Yang Zhou**[1☯], **Xiaofeng Fang**[2], **Jiang Liu**[1], **Guiyu Qu**[1]*

**1** School of Nursing, Weifang Medical University, Weifang, China, **2** Weifang People's Hospital, Weifang, China

☯ These authors contributed equally to this work.
* qugy@wfmc.edu.cn

**Data Availability Statement:** All relevant data are within the paper and its Supporting information files.

## Abstract

### Objective

The aim was to evaluate the effectiveness and safety of the vapocoolants for vascular puncture in children and adults.

### Method

The search was carried out in PubMed, Web of Science, Embase and The Cochrane Library, from inception to March 2022. Randomized controlled trials comparing vapocoolants to control conditions for participants received intravenous cannulation or arterial puncture were included. Two reviewers independently performed selection of studies, data extraction, and assessment of risk of bias. The analysis was performed using fixed or random-effects model with mean differences or standardized mean difference and risk ratios.

### Results

A total of 25 studies involving 3143 participants were included. Compared with control conditions, vapocoolants may not decrease the pain of patients with arterial puncture (SMD = -0.36, 95% CI = -0.92 to -0.19, $P$ = 0.20), but may more effectively relieve pain for adults received vein puncture (SMD = -0.65, 95% CI = -0.85 to -0.45, $P$ < 0.00001). The application of vapocoolant increased the procedural difficulty of medical personnel (RR = 2.49, 95% CI = 1.62 to 3.84, $P$<0.000 1) and participants were more willing to use the spray in the future (RR = 1.88, 95% CI = 1.34 to 2.64, $P$ = 0.0002). There was no significant difference for the first attempt success rate of the procedure and the occurrence of adverse events.

### Conclusions

Vapocoolant spray may relieve pain in adults received vein puncture and cannot cause severe side effects, but is ineffective in children. It also had no effect on patients with arterial puncture. In addition, the application of spray increases procedural difficulties for medical professionals, but does not decrease first attempt success rate, and many patients would

**Funding:** The author(s) received no specific funding for this work.

**Competing interests:** The authors have declared that no competing interests exist.

like to use the spray again for pain relief in the future. Thus, more rigorous and large-scale studies are needed to determine its effectiveness in vascular access.

## Introduction

Vascular access is a common invasive procedure in hospitals, especially intravenous catheters, which is particularly important for prevention and treatment. This procedure is painful, which might increase the anxiety levels in patients [1] and may lead to development of needle fear [2]. A meta-analysis [1] reported that needle fear was exhibited mostly in children (60%-100%), 20–50% in adolescents and 20–30% in adults. Existing studies have demonstrated that these negative experiences of needle puncture can not only increase the experience of pain through psychological and physiological mechanisms [3–5], but can also decrease patient compliance and make it more difficult for medical personnel to perform the procedure. In severe cases, there may be some consequences, including refusing to puncture and more general health care avoidance [1, 6–8]. These showed the importance of pain management. Pain is one of the major challenges of public health with an enormous impact on both individuals and society [9] and multiple clinical studies have reported that failure to treat pain can cause both short and long-term complications [10, 11]. Therefore, it is essential to find an effective intervention for pain relief of vascular access.

The most common technique for relieving pain in vascular access is a local anesthetic, but lidocaine injections are cumbersome and injection in itself can be painful [12]. Previous studies have also suggested that topical anesthesia provided comparable pain control [13, 14]. EMLA has been applied as an alternative tool for pain reduction. Unfortunately, there are some factors leading to infrequent use of topic local anesthesia. More specifically, it takes considerable time to produce a sufficient effect [14–16], higher cost [17], and may cause allergic contact dermatitis [18, 19].

Based on this situation, vapocoolant sprays seem to be a better anesthetic since it is a non-invasive operation, low cost, widely available and achieves skin anesthesia quickly [20–22]. Moreover, some studies had proven the effectiveness of vapocoolant sprays in immunization injection [23, 24]. In 2014, Hogan et al. [25] published a systematic review of vapocoolants for reducing pain from venipuncture and venous cannulation in children and adults, found that vapocoolants was effective in adults with venipuncture but not in children. However, a recent study [26] revealed contradictory results, that was, vapocoolants was effective no matter whether children or adults with intravenous cannulation. Since the previous meta-analysis has been conducted for nearly a decade, it is necessary to conduct a new meta-analysis on relevant high-quality studies again to explore the effect of vapocoolants for different types of subjects. Besides, many studies have also explored the effect of sprays on arteries, but there remains considerable controversy regarding the effectiveness of spray for arterial catheters. Therefore, we aim to systematically meta-analysis the study of intervention in vapocoolants and assess the effectiveness and safety of the vapocoolants to provide reliable and convincing evidence.

## Materials and methods

This review was performed according to Preferred Reporting Items for Systematic Reviews and Meta-Analyses Statement (PRISMA) guideline [27], and the methodology of this systematic review and meta-analysis has published in Prospero Platform (CRD42022326470).

## Search strategies

The search was carried out in the following databases: PubMed, Web of Science, Embase and Cochrane Library, from inception to March, 2022. The search strategy applied MeSH terms and keywords relating to vapocoolant spray, intravenous cannulation or arterial puncture. Detailed retrieval steps for PubMed can be available in supplementary material. In addition to the database search, reference lists of all included articles were manually screened to identify any additional eligible studies. Details of the search strategies were shown in S1 Table.

## Eligibility criteria

The criteria for research selection based on the PICO principles are as follows: (1) participants (P), children and adults who required intravenous cannulation or arterial puncture. (2) intervention (I) was vapocoolant spray. (3) comparison (C) included usual care, placebo spray or no intervention. (4) outcome measures (O) were assessed the pain intensity, adverse effect, the first attempt success rate, procedural difficulty and patient intentions. The primary outcome was pain scores after puncture which was assessed by a pain numeric rating scale (NRS) or visual analogue scale (VAS). We included all randomized controlled trials (RCTs) and clinical trials. Furthermore, the articles had to be published in English. Studies would be included if they report one of the above outcome measures. Retrospective or prospective cohort studies, review articles, case reports, conference abstracts, trial protocols and original studies without full texts were likewise excluded.

## Study selection

Study selection was conducted in EndNoteX9.1. Two reviewers (LW and YZ) independently screened the articles based on titles and abstracts after removing duplicate studies, and then the full text should be screened according to the inclusion criteria. Any arguments generated in the above process were solved by the two reviewers through negotiation, otherwise, needed to be handled by the third reviewer (XFF).

## Data extraction and quality assessment

Data was extracted independently by two reviewers using the predetermined Excel sheet and the pilot survey was performed on included three literatures to ensure consistency and relevancy of the extracted information. The sheet was extracted study characteristics, such as study design, sample size, first author, publication year, country, vapocoolant type, cannula size, spraying distance, and spraying time. The Cochrane Collaboration Tool [28] was used to explore the sources of bias in the included randomized trials. We evaluated included studies of six aspects (assessing selection bias, performance bias, detection bias, attrition bias, reporting bias, other bias) and each item was categorised as high risk, low risk, or unclear risk. As mentioned, inter-reviewer disagreements regarding the methodological quality of the included literatures were initially discussed by LW and YZ to reach a consensus or consulted the third reviewer (XFF).

## Statistical analyses

We conducted the Statistical analyses in Review Manager (RevMan) 5.1 software and Stata Statistical Software, and the fixed or random-effect model with a 95% confidence interval (CI) was used. For continuous variables, mean differences (MD) or standardized mean difference (SMD) were calculated. For dichotomous variables, risk ratios (RRs) were calculated. Inter-study heterogeneity was assessed by chi-squared test with a significance level of 0.1, and using

the $I^2$ statistic to quantify it. Sensitivity analysis was conducted to identify the influence of individual studies on the overall studies by omitting one study at a time. For heterogeneity analysis, subgroup meta-analysis and meta regression were performed to assess the level of heterogeneity. While publication bias was inspected by funnel plots and the asymmetry was evaluated by using Egger regression test. Some studies did not provide efficacy data of pain score, so we used methods recommended by Luo [29] and Wan [30] to calculate mean and standard deviation, which was procured by LW with the supervision of another reviewer (YZ).

## Results

### Search results

An initial pool of 4576 literatures were located from three databases, 769 studies were deleted due to duplication, and then 3807 articles needed to be screened titles and abstracts. 49 articles were selected for full-text review, and 25 of the studies were deemed eligible for inclusion according to inclusion criteria in the review. The process of studies identification and selection is presented in Fig 1.

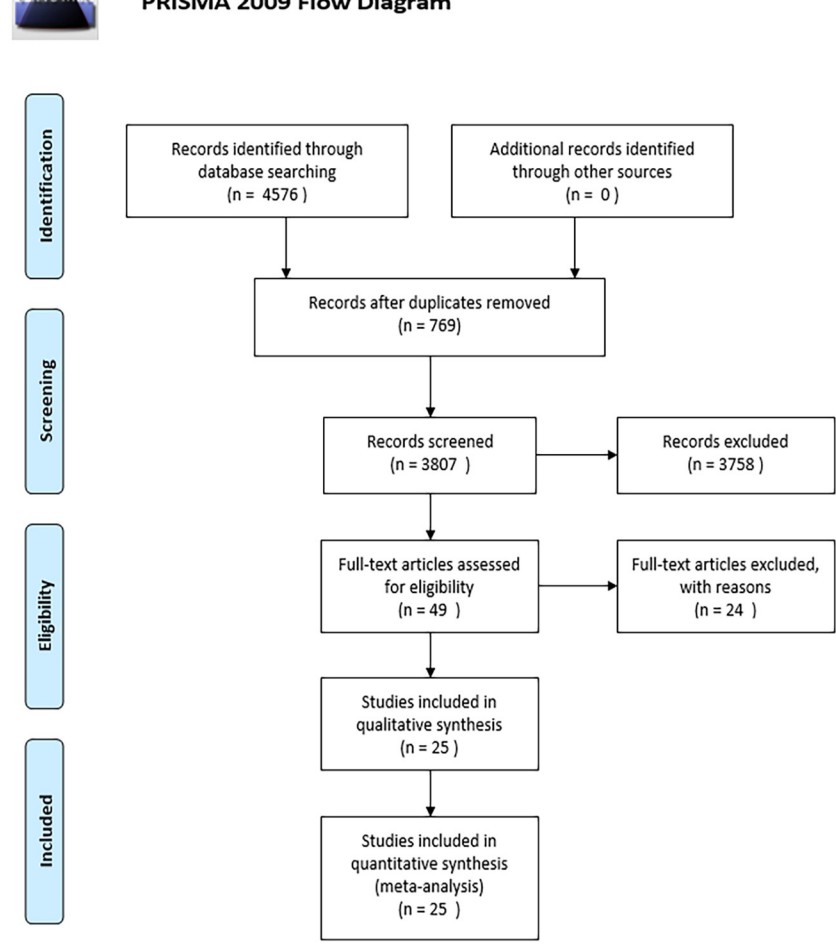

**Fig 1. Study selection.** *From*: Moher D, Liberati A, Tetzlaff J, Altman DG, The PRISMA Group (2009). *Preferred Reporting Items for Systematic Reviews and Meta-Analyses: The PRISMA Statement. PLoS Med 6(7): e1000097. doi:10.1371/journal.pmed1000097* **For more information, visit** www.prisma-statement.org.

## Study characteristics

25 RCTs were included in this review (Table 1) and all were issued in journals, except for one clinical trial result [31]. 3 studies were published before 2000 (in 1990 [32] 1995 [33] and 1999 [34], 7 studies were published between 2001–2010 [35–41], and the remaining 15 studies were published in 2011 and 2022 [31, 42–55]. 11 of the included studies originated from USA [31, 34–36, 40, 43–45, 47, 49, 51], 3 each from UK [32, 33, 39] and Iran [46, 48, 55], 2 each from Australia [37, 41] and Turkey [42, 53], and one each from Thailand [54], Germany [50], India [52] and Canadia [38]. The participants of study in 19 were adults [31–34, 37, 39–42, 44, 45, 47–54], 6 [35, 36, 38, 43, 46, 55] were children.

The intervention of all the studies was venipuncture and intravenous cannulation, except for 3 were arterial puncture [39, 48, 52]. Regarding the type of vapocoolant in the experimental groups, ethyl chloride was implemented in 12 studies [31–37, 39, 42, 44, 52, 54], fluorohydro-carbon was conducted in 7 studies [38, 40, 43, 45, 47, 49, 51], 2 studies were alkane mixture [41, 48] and 4 were not described [46, 50, 53, 55]. 25 studies compared vapocoolant to non-anaesthesia, either directly no-treatment [32–34, 37, 39, 46, 53, 55] or amongst other arms including placebos [31, 34, 38, 41, 42, 44, 45, 47–49, 51, 54], ice pack [43, 52] and isopropyl alcohol [35, 36, 50].

All participants were rated their degree of pain perception after cannulation by different scales. These tools were previously testified as reliable methods in measuring pain and discomfort. 10 included studies used pain numeric rating scale (NRS), 15 studies used a visual analogue scale (VAS). Furthermore, 21 studies [31–34, 36–46, 48, 50, 52–55] reported pain outcome with continuous variables, of which 3 studies [35, 41, 43] expressed outcome by interquartile range and then we applied methods recommended by Luo and Wan to calculate mean and standard deviation. The remaining 4 studies [35, 47, 49, 51] were presented as dichotomous variables.

There were 15 studies reported safety outcomes. Among them, 7 studies [31, 36, 38, 42, 44, 45, 47] reported no side effects related to the use of vapocoolant spray or the control group in any subjects, while other studies reported various adverse reactions, including skin redness (n = 2) [41], skin blanching(n = 5) [32], erythema(n = 33) [34, 50], burning sensation(n = 1) [37], coolness and numbness(n = 1) [49] and cold feeling(n = 15) [37, 49]. Moreover, one study [48] did not report the number of patients experiencing adverse events.

## Risk of bias

Details of the risk of bias assessment were shown in Fig 2 and S2 Table. All studies were evaluated as having an unclear or high risk of bias in at least one domain, except for 4 studies were low risk [38, 45, 48, 49]. All of the studies claimed to be randomized, however, 9 studies did not explicitly mention the methods applied to perform random sequence [31, 33–35, 42–44, 47, 54]. 13 studies reported data material on adequate allocation [35–38, 40, 41, 45, 47–50, 52, 54], while others were considered unclear risk. It is noteworthy that only 9 of the included studies were described as double blinded [36, 38, 41, 44, 45, 47–49, 51]. 12 studies revealed the content of the blinding of the participants and personnel [31, 34, 36, 38, 41, 44, 45, 47–51], and 5 studies mentioned that the blind was not available to patients and personnel [40, 42, 46, 52, 55]. 13 studies have described for assessment by evaluators who were unaware of the intervention [36, 38, 41, 43–45, 47–49, 51–53, 55] and only 2 studies reported the blindness of the assessor [40, 46]. 6 studies had an unclear risk of incomplete outcome data [32–34, 41–43, 46] and the rest of studies mentioned the reasons and the number of exclusion participants. 13 studies have measured all the predefined outcomes [31, 38, 39, 41, 44, 45, 47–53, 55].

**Table 1. General description of studies included in the review.**

| Study | Sample | Study design | Surgery type | Spraying time | Spraying distance | cannula size | Vapocoolant type | Control group | Assessment |
|---|---|---|---|---|---|---|---|---|---|
| Mitryn [31] USA 2018 | N = 30 Adults | RCT 2 parallel groups | Intravenous Insertions | 4–10 s | 3–9 in | ND | Ethyl Chloride | Nature's Tears | Pain Willingness of the Patient to choose the allocated spray in the future Adverse effects |
| Armstrong [32] UK 1990 | N = 120 Adults | RCT 3 parallel groups | venepuncture | 10 s | 8 in | 20G | Ethyl chloride | No treatment | Pain Adverse effects |
| Selby [33] UK 1995 | N = 160 Adults | RCT 4 parallel groups | venous cannulation | 10 s | 20 cm | 20G | Ethyl chloride | No treatment | Pain Number of successes |
| Crecelius [34] USA 1999 | N = 88 Adults | RCT 2 parallel groups | Venous Cannulation | 10 s | ND | 20G,22G | Ethyl chloride | water spray | Pain Anxiety due to spray Adverse effects |
| Ramsook [35] USA 2001 | N = 222 Children | RCT 2 parallel groups | venipuncture | 5 s | 6 inches | ND | Ethyl chloride | Isopropyl alcohol | Success rate on first attempt Adverse effects Procedure difficulty Impairment of vein visibility |
| Costello [36] USA 2006 | N = 129 Children | RCT 3 parallel groups | Intravenous Cannulation | 5 s | ND | 22G | Ethyl chloride | Isopropyl alcohol | Pain Success rate on first attempt Adverse effects |
| Robinson [37] Australia 2007 | N = 290 Adults | RCT 4 parallel groups | Intravenous cannulation | 5–10 s | 15–20 cm | 21, 20, 18 or 16 G | Ethyl chloride | No treatment | Pain Adverse effects |
| Farion [38] Canadia 2008 | N = 80 Children | RCT 2 parallel groups | Intravenous cannulation | 4–10 s | 8–18 cm | 22 or 24 G | 1,1,1,3,3-pentafluoropropane and 1,1,1,2-tetrafluoroethane | Nature's Tears | Pain Success rate on first attempt Adverse effects Procedure difficulty |
| France [39] UK 2008 | N = 59 Adults | RCT 3 parallel groups | Arterial Puncture | ND | 10 cm | 23-G | Ethyl chloride | No treatment | Pain on arterial puncture Pain at 5 min |
| Hartstein [40] USA 2008 | N = 92 Adults | RCT 2 parallel groups | Intravenous cannulation | 2–4 s | 7–12 cm | 22 or 18G | 1,1,1,3,3,-pentaflouropropane and 1,1,1,2-tetrafluoroethane | Standard method | Pain Mean anxiety scores |
| Hijazi [41] Australia 2009 | N = 201 Adults | RCT 2 parallel groups | Intravenous cannulation | 2s | 12 cm | 22, 20 or 18 G | COLD Spray | Water spray | Pain Success rate of cannulation Willingness to use product on future subjects Unexpected events Discomfort from both sprays |
| Çelik [42] Turkey 2011 | N = 41 Adults | Crossover Study | Venipuncture | 2 s | 10 cm | 18-G | Ethyl chloride | Placebo cream | Pain Adverse events |
| Waterhouse [43] USA 2013 | N = 95 Children | RCT 2 parallel groups | Intravenous Catheter | 4–10 s | 8–18 cm | 24,22,20 or 18-G | 1,1,1,3,3-pentafluoropropane and 1,1,1,2-tetrafluoroethane | Ice | Pain Patient Satisfaction Subject would want again Procedure difficulty |

*(Continued)*

**Table 1.** (*Continued*)

| Study | Sample | Study design | Surgery type | Spraying time | Spraying distance | cannula size | Vapocoolant type | Control group | Assessment |
|-------|--------|--------------|--------------|---------------|-------------------|--------------|------------------|---------------|------------|
| Fossum [44] 2016 USA | N = 38 Adults | crossover trial | Venous Cannulation | 5–8 s | Not describe | 20-G | Ethyl chloride | Nature's Tears | Pain Adverse effects Success rate on first attempt at cannulation Future willingness to use agent on patient |
| Mace [45] USA 2016 | N = 100 Adults | RCT 2 parallel groups | Venipuncture | 4–10 s | 3–7 in | 21-G | 1,1,1,3,3-pentafluoropropane or 1,1,1,2-tetrafluoroethane | Nature's Tears | Pain Adverse effects Success rate on first attempt at cannulation Spray time Willingness to use product on future subjects |
| Dalvandi [46] Iran 2017 | N = 40 Children | Crossover trial | intravenous cannulation | 2 s | 10 cm | ND | ND | No treatment | Pain |
| Edwards [47] USA 2017 | N = 72 Adults | RCT 2 parallel groups | Peripheral Intravascular Access | 2 s | 4–5 in | ND | 1,1,1,3,3-pentafluoropropane or 1,1,1,2-tetrafluoroethane | Placebo | Success rate on first attempt at cannulation Willingness to use product on future subjects Adverse effects Anxiety |
| Farahmand [48] Iran 2017 | N = 80 Adults | RCT 2 parallel groups | Arterial Puncture | 5 s | 20 cm | ND | Alkane mixtures | water spray | Pain Spray application pain Number of tries Adverse effects |
| Mace [49] USA 2017 | N = 300 Adults | RCT 2 parallel groups | intravenous cannulation | 4–10 s | 3–7 in | 22,20 or 18-G | 1,1,1,3,3-pentafluoropropane and 1,1,1,2-tetrafluoroethane | Nature's Tears | Pain Adverse effects Success rate on first attempt at cannulation Willingness to use product on future subjects |
| Rusch [50] Germany 2017 | N = 450 Adults | RCT 4 parallel groups | Venous Cannulation | 2 s | 5 cm | 20 or 17-G | ND | alcohol | Pain Failed venipuncture attempts Process times Adverse effects |
| Barbour [51] USA 2017 | N = 100 Adults | RCT 2 parallel groups | Venipuncture | ND | ND | 21-G | 1,1,1,3,3-pentafluoropropane or 1,1,1,2-tetrafluoroethane | Nature's Tears | Success rate on first attempt at cannulation Willingness to use product on future subjects Patient satisfaction |
| Dhami [52] India 2020 | N = 60 Adults | RCT 2 parallel groups | Arterial Puncture | 3 s | 12–18 cm | 23-G | Ethyl Chloride | Ice pack | Pain Success of first attempt Haematoma (within 2h) Success at first attempt time taken for puncture |

(*Continued*)

**Table 1.** (Continued)

| Study | Sample | Study design | Surgery type | Spraying time | Spraying distance | cannula size | Vapocoolant type | Control group | Assessment |
|---|---|---|---|---|---|---|---|---|---|
| Basak [53] Turkey 2021 | N = 88 Adults | RCT 2 parallel groups | Venipuncture | 10–15 s | 25 cm | ND | ND | No treatment | Pain Blood/Injection Fear STAI |
| Supan [54] Thailand 2021 | N = 176 Adults | RCT 3 parallel groups | Intravenous Cannulation | 10s | 10 cm | 22-G | Ethyl Chloride | Mineral solution | Pain Success rate on first attempt at cannulation Patient Satisfaction |
| Ghasemi [55] Iran 2022 | N = 31 Children | RCT 2 parallel groups | venous cannulation | 5 s | 25 cm | 22-G | ND | No treatment | Pain |

RCT: randomized controlled trial, ND: not describe

## Meta-analysis

Meta-analyses were performed for the pain score after puncture, adverse effect, the first attempt success rate, procedural difficulty and patient intentions. In the following paragraphs, we present the results for effect sizes comparing the vapocoolant and control groups.

**Pain intensity.** *Pain scores after arterial puncture.* A total of 2 RCTs were included [48, 52], 140 adults and the outcomes were measured by the VAS scale. The results showed that there was no statistically significant difference between the vapocoolant spray group and the control group (MD = -0.36, 95% CI = -0.92 to -0.19, $P = 0.20$; $I^2 = 0\%$, $P = 0.39$; Fig 3).

*Pain scores after intravenous cannulation.* There are 19 [31–34, 36–46, 50, 53–55] studies which measured post-venipuncture pain score using continuous variables, but only 17 [31–34, 36, 38, 40–46, 50, 53–55] studies included in meta-Analysis. Two studies [37, 39] offered a 95% confidence interval in the VAS-score of the outcome measures and we cannot extract available data.

A total of 17 [31–34, 36, 38, 40–46, 50, 53–55] studies involving 1708 patients were included in this meta-analysis. Using the random effects model to synthesize pain score after intravenous cannulation assessed by different pain level measurement tools, however, this result was considerable heterogeneity (84%) and we performed a sensitivity analysis and found that the results were more stable after removing one study [46]. This may be related to the fact that this study recruited children with thalassemia who needed regular blood transfusions. Regular blood puncture can cause anxiety which makes children suffer from more pain than other people. After sensitivity analysis, the result indicated that vapocoolant spray had a significant effect on pain relief (SMD = -0.57, 95% CI = -0.76 to -0.39, $P < 0.00001$; $I^2 = 68\%$, $P < 0.0001$; Fig 4). Then the subgroup analysis was performed according to age, spray time, spray distance, vapocoolant type and cannula size.

**Subgroup analyses.** In the subgroup analysis of different age groups among patients, compared with adult participants, the vapocoolant spray was not statistically more efficacious than placebo spray/no intervention for children (SMD = −0.31, 95%CI = −0.66 to 0.04, $P = 0.08$; $I^2 = 54\%$, $P = 0.09$; Fig 5). For distance of vapocoolant spray, no matter whether the distance of vapocoolant spray was less than 10 centimeters, it was all effective for relieving pain. While in the studies of more than 10 centimeters, we consider that the result was stable and reliable since the heterogeneity of the results was very low (SMD = − 0.68, 95%CI = −0.91 to -0.44, $P < 0.00001$; $I^2 = 31\%$, $P = 0.21$; Fig 6). According to the classification of spray time,

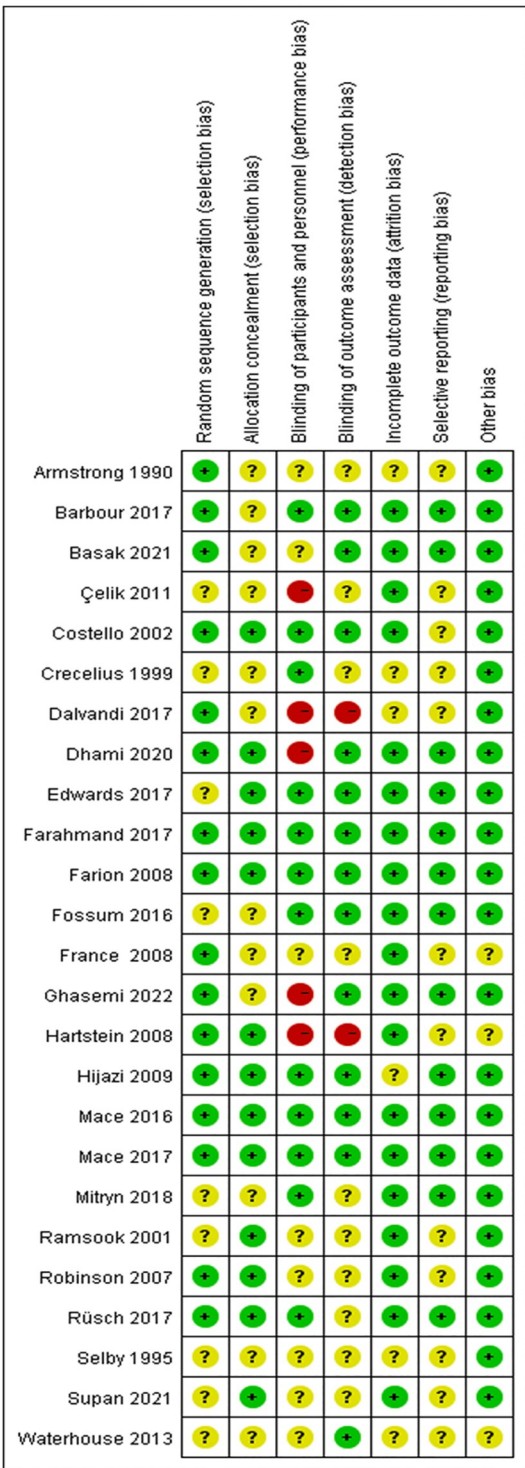

**Fig 2. Summary of risk of bias.**

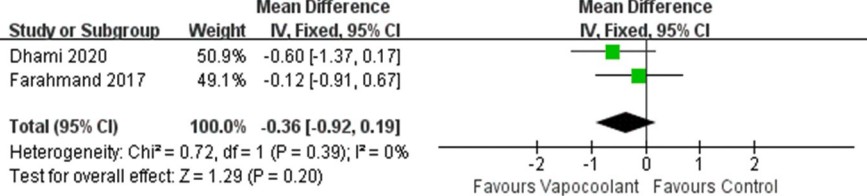

**Fig 3. Forest plot of pain scores during artery puncture.**

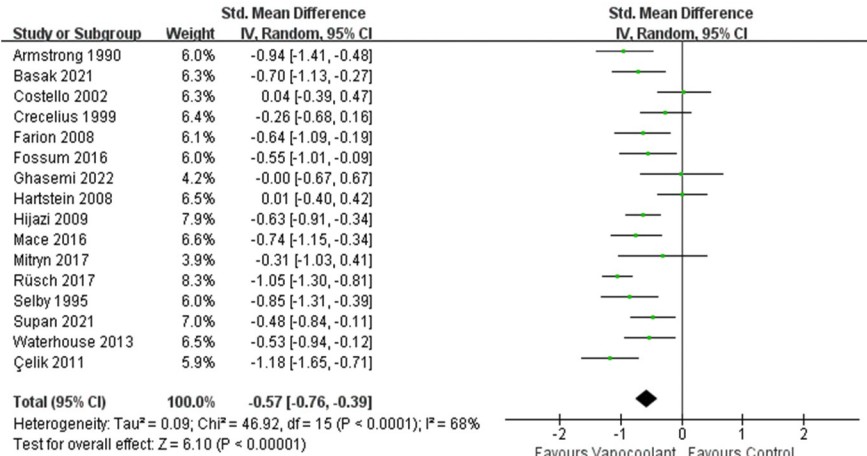

**Fig 4. Forest plot of pain scores during intravenous cannulation.**

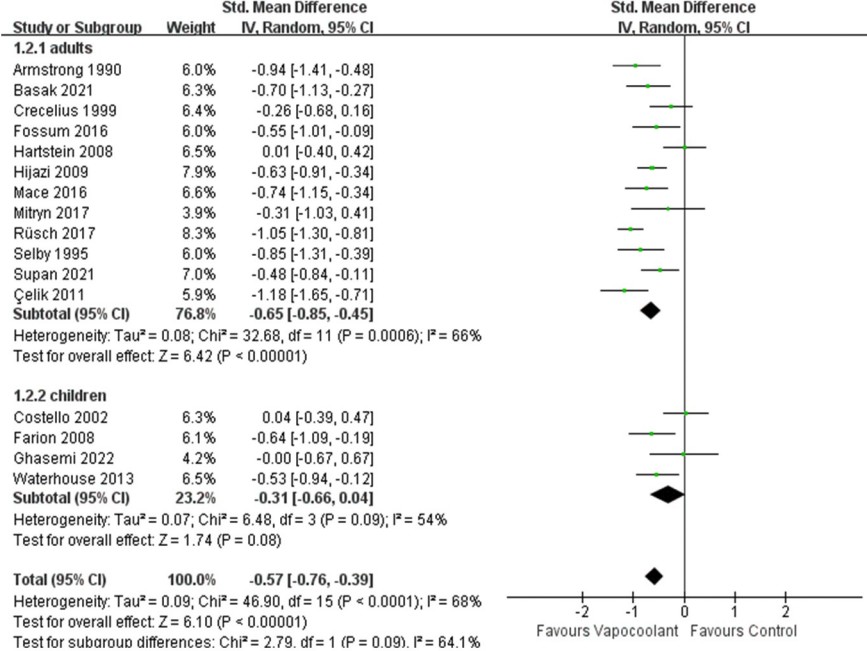

**Fig 5. Subgroup analysis of pain scores during intravenous cannulation: Vapocoolants in children vs adults.**

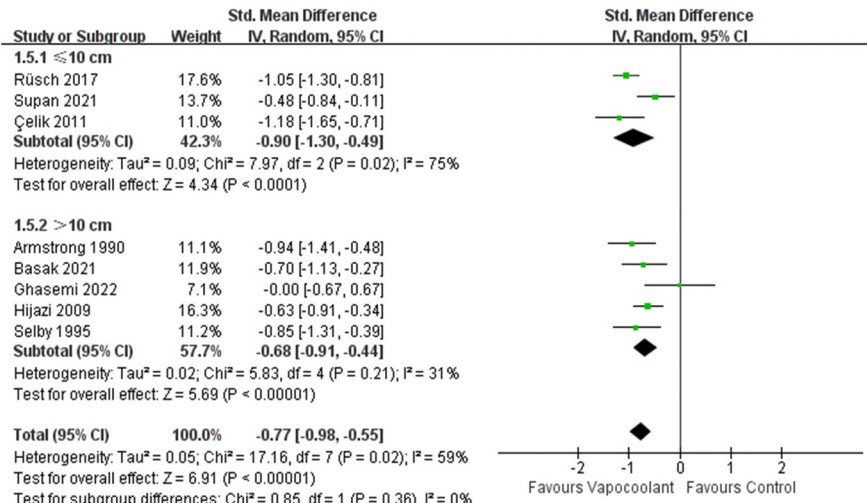

**Fig 6. Subgroup analysis of pain scores during intravenous cannulation: Vapocoolants spray distance ≤10cm vs >10cm.**

there was a no statistically significant subgroup difference (Chi$^2$ = 0.76, $P$ = 0.38, I$^2$ = 0%; Fig 7). Furthermore, the synthesized effect size of subgroup spray time more than 5 seconds (SMD = -0.66, Z = 7.81, $P$< 0.00001) was mildly larger than subgroup spray time less than 5 seconds (SMD = -0.48, Z = 2.59, $P$ = 0.01). For patients who received ethyl chloride (SMD = − 0.56, 95%CI = -0.81 to -0.32; I$^2$ = 62%, $P$ = 0.007; Fig 8) or other type of vapocoolant (SMD = − 0.51, 95%CI = -0.82 to -0.19; I$^2$ = 64%, $P$ = 0.04; Fig 8), vapocoolant group was more effective than the control group. A subgroup analysis could not be performed because cannula size was different or not specified. Meta-regression subgroup analyses were performed based on age, spray time, spray distance and vapocoolant type to determine the possible effects of these

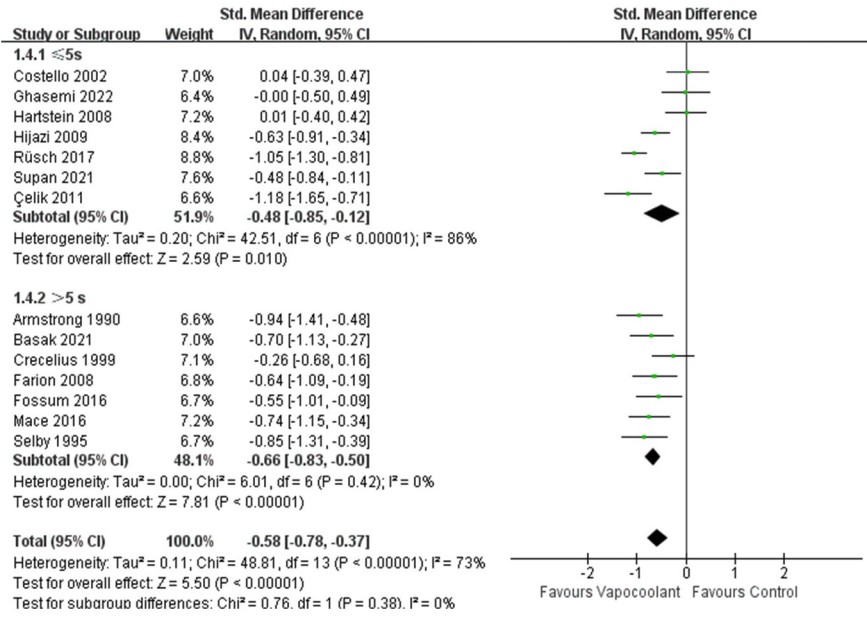

**Fig 7. Subgroup analysis of pain scores during intravenous cannulation: Vapocoolants spray time ≤5s vs >5s.**

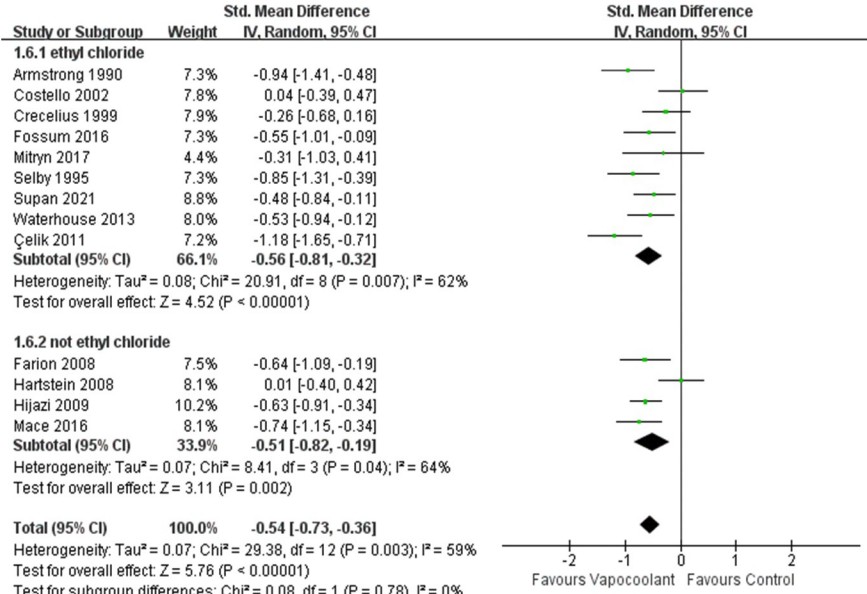

**Fig 8. Subgroup analysis of pain scores during intravenous cannulation: Vapocoolants type ethyl chloride vs not ethyl chloride.**

variables. The result indicated that spraying time ($P = 0.025$; S3 Table) and spraying distance ($P = 0.006$; S3 Table) of vapocoolant may cause high heterogeneity.

**Adverse effect.** 12 studies [31, 32, 34, 36–38, 41, 42, 44, 45, 47, 49] (n = 1293) were conducted in a meta-analysis and the result of the fixed effects model demonstrated that using the vapocoolant did not increase the risk of adverse effect (RR = 1.66, 95% CI = 0.91 to 3.03, $P = 0.10$; Fig 9), and had a low heterogeneity ($I^2 = 42\%$, $P = 0.14$).

**Patients attitude.** 898 patients in 7 studies [31, 41, 43, 45, 47, 49, 51] were analyzed to explore participants' future willingness to desire the same sprays. Compared with the control group, more patients would like to use the vapocoolant spray in the future, and this was statistically significant (RR = 1.88, 95% CI = 1.34 to 2.64, $P = 0.0002$; $I^2 = 88\%$; Fig 10).

**First attempt success rate.** 14 studies [33, 35, 36, 38, 41, 43–45, 47, 49–52, 54] including 1865 participants were synthesized to analyze the success rate of first puncture attempts in both groups. No significant negative effect of vapocoolant on first attempt success rate

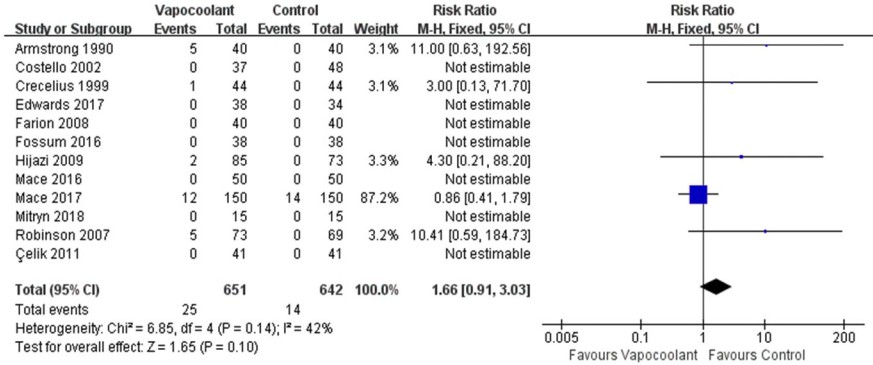

**Fig 9. Forest plot of adverse effect.**

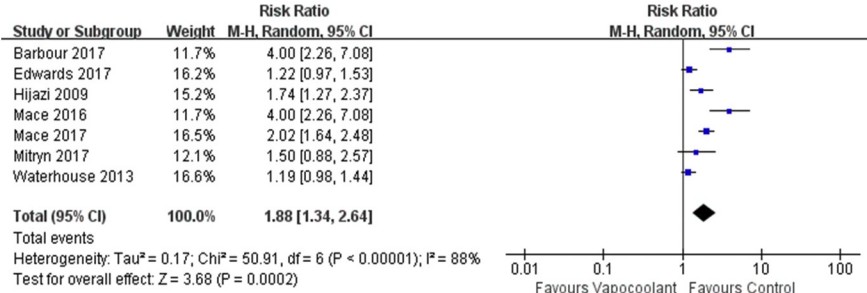

**Fig 10. Forest plot of patient's attitude.**

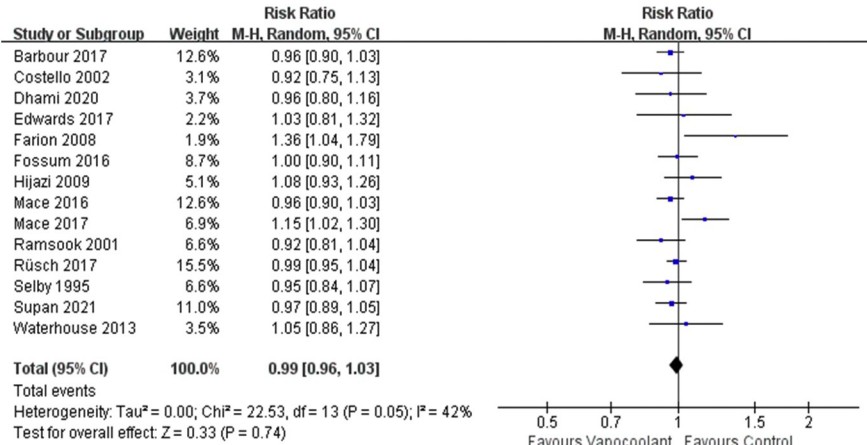

**Fig 11. Forest plot of first attempt success rate.**

compared with the control group was detected (RR = 0.99, 95% CI = 0.96 to 1.03, $P$ = 0.74; $I^2$ = 42%; Fig 11).

**Procedural difficulty.** 5 studies [35, 38, 43, 45, 51] were pooled in a meta-analysis to explore whether using of vapocoolants could increase the technical difficulty. However, we observed that the results were highly heterogeneous ($I^2$ = 72%). After sensitivity analysis, the result was stable after excluding Farion's article [38], indicating that the application of vapocoolants would increase the technical difficulty of medical staff (RR = 2.49 95% CI = 1.62 to 3.84, $P$<0.0001; $I^2$ = 29%; Fig 12).

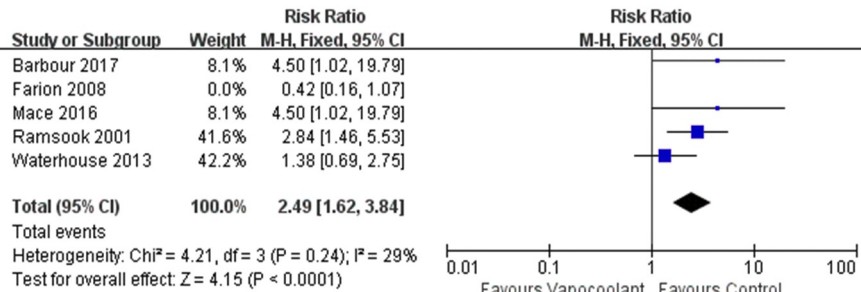

**Fig 12. Forest plot of procedural difficulty.**

**Publication bias.** The Egger test was conducted to estimate publication bias when 10 or more included studies, the results indicated no obvious publication bias (See to S4 Table). Publication bias analysis was not performed because only 6 studies of the adverse events were valid.

## Discussion

This meta-analysis is designed to assess the efficacy and safety of vapocoolant spray for children and adults by reviewing and analyzing of the existing studies. We incorporated data from 25 RCTs to evaluate the effect of vapocoolants on pain intensity, adverse events, the first attempt success rate, procedural difficulty and patient attitude. The result suggested that using vapocoolant spray can significantly decrease pain scores of subjects after venipuncture. However, there was a high degree of heterogeneity and we suspected it might be related to the population. Subgroup analysis of the population indicated that vapocoolant spray has an active efficacy in the treatment of decreasing the pain of venipuncture and Venous Cannulation among adult participants. But children's studies did not demonstrate a reduction in pain with vapocoolant spray and had no statistically significant subgroup difference. This conclusion is consistent with a previous meta-analysis [25] that evaluated the effects of vapocoolant spray. Notably, one study suspected that vapocoolants are ineffective in children because they perceive cold feeling as a discomfort [56], and the vapocoolant could increase the feeling of cold.

Rusch et al. [57] found that vapocoolant spray significantly mitigated discomfort compared to lidocaine after evaluating the overall discomfort associated with both the anesthetic application and artery puncture. A meta-analysis also stated that cryotherapy emerged as beneficial in reducing patients' perception of arterial puncture-related pain [58]. Nevertheless, two studies [48, 52] comprising 140 patients were included in our meta-analysis and reported that vapocoolant was not effective in reducing the intensity of arterial puncture pain compared to the control group. Moreover, the same finding was also revealed in a study including 59 participants over the age of 16 years received arterial puncture [39]. This effect might be because of the fact that more nociceptors were activated during arterial puncture [12, 48], whereas cold sprays only caused peripheral vasoconstriction but cannot relieve the pain of deep receptors. Therefore, the effectiveness of spray for arterial puncture remains uncertain. Moreover, although 80% of people currently consider arterial punctures to be very painful [59], most have not undergone analgesic techniques [58], which also reminds us to focus on pain management for patients with arterial puncture.

Of note, none of the included studies reported serious adverse events and all reported adverse reactions were short-term and mild. We also found that the first attempt success rate was not related to vapocoolant spray. In addition, in contrast to Griffith [60] findings, the application of vapocoolant increased the procedural difficulty of medical personnel. Improper spraying or delay in performing the procedure may reduce the efficacy of vapocoolant, and it may be difficult to perform the given procedure in this short period of time [45]. Furthermore, participants were more willing to use the vapocoolant spray in the future, which also reflected the satisfaction of patients with spray application.

The results of other subgroup analysis found that, when spraying time exceeds 5 seconds, vapocoolants had a better effect in relieving pain. It is consistent with Rao [61] that vapocoolant sprayed twice and each over 5 seconds effectively reduced pain on venipuncture, but local cells may die due to severe local hypothermia if sprayed exceeds 10 seconds [38]. Because most studies were not provided sufficient details of cannula size, subgroup analysis could not be conducted. One study [50] showed that there were more advantages in cryopanaesthesia compared to lidocaine infiltration when using large venous cannulas (17G or larger). Also, in

contrast to the previous meta, our study showed that the spray of distance and time contributed to a high degree of heterogeneity. Therefore, we need to further explore the effects of time, distance, and cannula size on the effectiveness of the spray in the future.

With an aging population and multiple diseases, more and more people need to be treated, and the pain of vascular access has become an increasingly common and important public health problem [62, 63]. Vapocoolant spray is a well-tolerated and convenient method, immediate onset of effect, an affordable and environmentally friendly alternative to pain-reducing anesthetic. It suppresses pain signals by spraying a volatile liquid on the surface of the skin that rapidly evaporates causing a drop in temperature and a decrease in nerve conduction velocity [20–22, 64]. Although our results show no effect in children undergoing venipuncture and no effect in patients with arterial puncture, but there is no doubt that it is an excellent pain reliever for venipuncture in adults. Therefore, we still recommend following the principle of vapocoolant analgesia in future clinical practice.

## Limitations

To the best of our acknowledge, we first assessed original studies of arterial puncture in the meta-analysis. Unfortunately, there are no sufficient randomized controlled trials that specifically rated the effects of vapocoolant spray on arterial puncture, and some published studies have not reported adverse events and procedural difficulty. Therefore, it may not really reflect the effects of the intervention. Moreover, the existing research design is still not rigorous. First, most original studies used the spray at different distances and time. Secondly, rarely consideration has been given to the effect of cannula size in previous studies. Thirdly, not all studies were blinded. There are a few high-quality literatures, so we have included all relevant studies, and more high-quality clinical studies are needed to support our research results in the future. Furthermore, some data is obtained through transformation, which may introduce bias.

## Conclusions

The use of vapocoolant spray may relieve pain in adults received vein puncture and the spray can not cause severe side effects, but is ineffective in children. It also had no effect on patients with arterial puncture. In addition, the application of spray increases procedural difficulties for medical professionals, but does not decrease first attempt success rate, and many patients would like to use the spray again for pain relief in the future. More rigorous and large-scale studies are needed to determine its effectiveness in vascular access.

## Supporting information

**S1 Table. Search strategy.**
(DOCX)

**S2 Table. Quality assessment of studies included.**
(DOCX)

**S3 Table. Summary results of regression analysis.**
(DOCX)

**S4 Table. Assessment of publication bias.**
(DOCX)

**S1 Checklist. Completed PRISMA checklist for the study.**
(DOC)

**S1 Data.**
(XLSX)

## Author Contributions

**Conceptualization:** Lan Wang, Yang Zhou.

**Data curation:** Lan Wang, Liu Fang, Yang Zhou, Xiaofeng Fang.

**Formal analysis:** Lan Wang, Liu Fang.

**Investigation:** Liu Fang.

**Methodology:** Lan Wang, Yang Zhou.

**Software:** Lan Wang, Liu Fang, Yang Zhou, Jiang Liu.

**Supervision:** Guiyu Qu.

**Validation:** Liu Fang.

**Writing – original draft:** Lan Wang, Liu Fang, Yang Zhou.

**Writing – review & editing:** Liu Fang.

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
