## [Decision Letter · Decision Letter 0]

2 Oct 2022

PONE-D-22-20412Efficacy and safety of vapocoolant spray for vascular puncture in children and adults: a systematic review and meta-analysisPLOS ONE

Dear Dr. Qu,

Thank you for submitting your manuscript to PLOS ONE. After careful consideration, we feel that it has merit but does not fully meet PLOS ONE’s publication criteria as it currently stands. Therefore, we invite you to submit a revised version of the manuscript that addresses the points raised during the review process.

We look forward to receiving your revised manuscript.

Kind regards,

Tariq Jamal Siddiqi

Academic Editor

PLOS ONE

Journal Requirements:

"NO"

“There was no competing interests.”

Reviewers' comments:

Reviewer's Responses to Questions

**Comments to the Author**

1. Is the manuscript technically sound, and do the data support the conclusions?

Reviewer #1: Yes

2. Has the statistical analysis been performed appropriately and rigorously? 

Reviewer #1: Yes

3. Have the authors made all data underlying the findings in their manuscript fully available?

Reviewer #1: Yes

4. Is the manuscript presented in an intelligible fashion and written in standard English?

Reviewer #1: Yes

5. Review Comments to the Author

Reviewer #1: Wang et al. conducted a meta-analysis on “Efficacy and safety of vapocoolant spray for vascular puncture in children and adults”, and found that vapocoolant spray was effective in reducing pain from vein puncture in adults, carried no significant adverse effects, and had a positive impact on patient’s attitude towards future use of the spray. However, it was not found to be effective in reducing pain in children, in arterial puncture, and increased the difficulty of the vascular access procedures. In my opinion, certain edits are still required:

1. In the abstract, the authors need to mention that the analysis was performed using random-effects model with risk ratios and mean differences.

2. In line 53-54 please replace the phrase “This procedure is painful, easy to increase….” With “This procedure is painful, which might increase the anxiety levels in patients and may lead to development of needle fear”.

3. In line 55 the percentage of prevalence of needle fear among children can be mentioned.

4. Authors should rephrase line 56-57 to make better sense of what they are trying to depict.

5. The paragraphs under eligibility criteria and study selection could be merged under one heading.

6. For the outcome pain intensity, authors need to pool effect sizes from all possible studies irrespective of the risk of bias an individual study carries. To enhance the reliability of the result, a sub-group analysis could be performed comparing low or moderate risk studies with high risk studies.

7. In the discussion section authors should talk how their findings are relevant and what do they add to the existing literature. Clinical implications also need to be included.

8. Line 362-368 should be mentioned in the limitations section.

9. The outcome procedural difficulty is an integral part of the meta-analysis and should be mentioned in the conclusion section.

6. PLOS authors have the option to publish the peer review history of their article (what does this mean?). If published, this will include your full peer review and any attached files.

Reviewer #1: No

---

## [Author Response · Author response to Decision Letter 0]

2 Nov 2022

Dear Editor,

Thank you very much for giving us an opportunity to revise our manuscript. We appreciate editor and reviewers very much for their positive and constructive comments and suggestions on our manuscript entitled “Efficacy and safety of vapocoolant spray for vascular puncture in children and adults: a systematic review and meta-analysis” (ID: PONE-D-22-20412). 

A point-by-point response to the reviewer's comments, a tracked version of the manuscript using the 'Track Changes' function in MS Word, and a clean version of the manuscript with all changes accepted are all submitted. Meanwhile, we checked the whole article carefully to revise possible editing and punctuation mistakes. We hope you are satisfied with our revised manuscript. Our replies to the comments of the reviewers are attached below.

At the same time, we ensure that our manuscript meets PLOS ONE's style requirements. At your suggestion we have made detail and repeated revisions for possible linguistic errors. The corresponding authors’ ORCID ID is 0000-0002-4281-2985, we have updated it in ‘Update my Information’. We have amended statements in the cover letter about funding, Data Availability and competing Interests. 

We would like to express our great appreciation to you and reviewers for comments on our paper. Looking forward to hearing from you.

Thank you and best regards.

Your sincerely,

Corresponding author

Quiyu Qu

Our Reply to the reviewer's Comments:

Reviewer#1' Comments to Author: 

Wang et al. conducted a meta-analysis on “Efficacy and safety of vapocoolant spray for vascular puncture in children and adults”, and found that vapocoolant spray was effective in reducing pain from vein puncture in adults, carried no significant adverse effects, and had a positive impact on patient’s attitude towards future use of the spray. However, it was not found to be effective in reducing pain in children, in arterial puncture, and increased the difficulty of the vascular access procedures. In my opinion, certain edits are still required

Dear Reviewer,

Thank you very much for your time spent in reviewing our manuscript and for your encouraging comments on its merits. After careful consideration, we have further revised the article. We hope that you will be more satisfied with the revised version.

1. In the abstract, the authors need to mention that the analysis was performed using random-effects model with risk ratios and mean differences.

Reply from authors: Thank you very much for your valuable suggestion. We agree with you very much. We have added the information in the manuscript. 

[Relative revision can be found in the abstract part, line 34-36]

2. In line 53-54 please replace the phrase “This procedure is painful, easy to increase….” With “This procedure is painful, which might increase the anxiety levels in patients and may lead to development of needle fear”.

Reply from authors: Thank you for your good advice. Your suggestions really mean a lot to us. According to your suggestion, we have replaced the phrase. Thank you very much.

[Relative revision can be found in the Induction part, line 59-60]

3. In line 55 the percentage of prevalence of needle fear among children can be mentioned.

Reply from authors: Thank you very much for your valuable suggestion. This suggestion is very important for us, and we have added the percentage of prevalence of needle fear among children.

[Relative revision can be found in the Induction part, line 61]

4. Authors should rephrase line 56-57 to make better sense of what they are trying to depict.

Reply from authors: Thanks for your valuable comments. It's a great help for us to have your suggestions. Through consideration and literature review, we modified the problem in the revised manuscript: Existing studies have demonstrated that these negative experiences of needle puncture can not only increase the experience of pain through psychological and physiological mechanisms [3-5], but can also decrease patient compliance and make it more difficult for medical personnel to perform the procedure. Thank you very much.

[Relative revision can be found in the Induction part, line 62-65]

5. The paragraphs under eligibility criteria and study selection could be merged under one heading.

Reply from authors: Thank you very much for your valuable suggestion. We have merged Data extraction and quality assessment under one heading.

[Relative revision can be found in the Materials and Methods part, line 128-140]

6. For the outcome pain intensity, authors need to pool effect sizes from all possible studies irrespective of the risk of bias an individual study carries. To enhance the reliability of the result, a sub-group analysis could be performed comparing low or moderate risk studies with high risk studies.

Reply from authors: Thank you very much for your valuable suggestion. This suggestion is very important for us. We have examined and revised the section of Pain scores after Intravenous cannulation. Thank you very much again for your valuable suggestion and understanding.

[Relative revision can be found in the Materials and Methods part, line 233-249]

7. In the discussion section authors should talk how their findings are relevant and what do they add to the existing literature. Clinical implications also need to be included.

Reply from authors: Thanks for your valuable comments. It's a great help for us to have your suggestions. We have added the information in the manuscript. Thank you very much.

[Relative revision can be found in the Discussion part, line 353-356, 372-375]

8. Line 362-368 should be mentioned in the limitations section.

Reply from authors: Thank you very much for your valuable suggestion. We have amended this section to the limitations section. 

[Relative revision can be found in the limitations part, line 392-395]

9. The outcome procedural difficulty is an integral part of the meta-analysis and should be mentioned in the conclusion section.

Reply from authors: Thank you very much for your valuable suggestion. We have added the information in the manuscript.

[Relative revision can be found in the Conclusions part, line 402-405]

---

## [Decision Letter · Decision Letter 1]

7 Dec 2022

Efficacy and safety of vapocoolant spray for vascular puncture in children and adults: a systematic review and meta-analysis

PONE-D-22-20412R1

Dear Dr. Qu,

We’re pleased to inform you that your manuscript has been judged scientifically suitable for publication and will be formally accepted for publication once it meets all outstanding technical requirements.

Kind regards,

Tariq Jamal Siddiqi

Academic Editor

PLOS ONE

Additional Editor Comments (optional):

Reviewers' comments:

Reviewer's Responses to Questions

**Comments to the Author**

1. If the authors have adequately addressed your comments raised in a previous round of review and you feel that this manuscript is now acceptable for publication, you may indicate that here to bypass the “Comments to the Author” section, enter your conflict of interest statement in the “Confidential to Editor” section, and submit your "Accept" recommendation.

Reviewer #1: All comments have been addressed

2. Is the manuscript technically sound, and do the data support the conclusions?

Reviewer #1: Yes

3. Has the statistical analysis been performed appropriately and rigorously? 

Reviewer #1: Yes

4. Have the authors made all data underlying the findings in their manuscript fully available?

Reviewer #1: Yes

5. Is the manuscript presented in an intelligible fashion and written in standard English?

Reviewer #1: Yes

6. Review Comments to the Author

Reviewer #1: (No Response)

7. PLOS authors have the option to publish the peer review history of their article (what does this mean?). If published, this will include your full peer review and any attached files.

Reviewer #1: No

---

## [Editor Report · Acceptance letter]

3 Feb 2023

PONE-D-22-20412R1 

Efficacy and safety of vapocoolant spray for vascular puncture in children and adults: a systematic review and meta-analysis 

Dear Dr. Qu:

I'm pleased to inform you that your manuscript has been deemed suitable for publication in PLOS ONE. Congratulations! Your manuscript is now with our production department. 

Kind regards, 

on behalf of

Dr. Tariq Jamal Siddiqi 

Academic Editor

PLOS ONE